# The Effect of Milk-Derived Extracellular Vesicles on Intestinal Epithelial Cell Proliferation

**DOI:** 10.3390/ijms252413519

**Published:** 2024-12-17

**Authors:** Shimon Reif, Liron Birimberg-Schwartz, Myriam Grunewald, Deborah Duran, Naama Sebbag-Sznajder, Tirtsa Toledano, Mirit Musseri, Regina Golan-Gerstl

**Affiliations:** 1Department of Pediatrics, Hadassah-Hebrew University Medical Center, Jerusalem 9166100, Israel; reif@hadassah.org.il (S.R.);; 2Hadassah Organoid Center, The Hadassah Medical Organization, Jerusalem 9166100, Israel; lbschwartz@hadassah.org.il (L.B.-S.);; 3Department of Pediatric Gastroenterology, Hadassah Medical Organization and Faculty of Medicine, Hebrew University of Jerusalem, Jerusalem 9166100, Israel; 4Department of Developmental Biology and Cancer Research, Faculty of Medicine, Hebrew University of Jerusalem, Jerusalem 9166100, Israel

**Keywords:** milk-derived extracellular vesicles, intestinal epithelial cells, β-catenin, barrier function

## Abstract

Inflammatory bowel disease (IBD) is a chronic, relapsing inflammation disorder of the gastrointestinal tract characterized by disrupted intestinal epithelial barrier function. Despite advances in treatment, including biological agents, achieving sustained remission remains challenging for many patients with IBD. This highlights the urgent need for novel therapeutic strategies. Milk-derived extracellular vesicles (MDEs) have emerged as a promising therapeutic option. In this study, we isolated and characterized MDEs and evaluated their effects on the function of intestinal epithelial cells (IECs). Using a murine model of Dextran Sulfate Sodium (DSS)-induced colitis, we observed that MDEs significantly ameliorated disease symptoms. The upregulation of β-catenin, a crucial mediator of Wnt signaling, in colonic tissues suggests that MDEs may facilitate epithelial regeneration and restore barrier function. In patient-derived colon organoids (PDCOs), MDEs were internalized and modulated the expression of key signaling molecules, such as the upregulation of β-catenin, cyclin D1, and the proliferation marker Ki67, indicating their potential to promote IEC proliferation and intestinal barrier repair. Importantly, MDEs demonstrated selective activity by downregulating β-catenin and cyclin D1 in colon cancer cells, leading to reduced proliferation. This selectivity indicates a dual therapeutic potential of MDEs for promoting healthy IEC proliferation while potentially mitigating malignancy risks.

## 1. Introduction

Inflammatory bowel disease (IBD) is a chronic inflammatory disease of the gastrointestinal tract that affects millions worldwide [1,2]. It manifests through debilitating symptoms, including severe, and sometimes bloody, diarrhea, abdominal pain, fatigue, and weight loss. While the pathogenesis of IBD is thought to be multifactorial, the disruption of the intestinal epithelial barrier is a common disease marker. In fact, intestinal epithelial cells (IECs) have an important impact on the pathophysiology of intestinal diseases, such as IBD, which are characterized by increased gut–blood barrier permeability, often referred to as a ‘leaky gut’. Leaky gut has been shown to precede the onset of IBD years before the inflammatory cascade begins.

Despite significant progress in IBD therapy, including biological therapies, many patients still face disease relapses and incomplete remission, underscoring the need for new therapeutic strategies. Current treatments often fail to address the root cause of epithelial barrier dysfunction, leaving a critical gap in long-term disease management. Extracellular vesicles (EVs), including milk-derived exosomes (MDEs), represent a novel therapeutic modality that could address these unmet needs.

Extracellular vesicles are nano-sized particles naturally secreted by cells, carrying bioactive molecules such as microRNA (miRNA), DNA, RNA, proteins, and lipids [3]. We have demonstrated that both human and cow MDEs are taken up by IECs, inducing the proliferation and differentiation of IECs in vitro [4] and exerting therapeutic and anti-inflammatory effects in a murine colitis model [5]. We further showed that MDE treatment affects several complementary pathways, including miRNA and regulation of the expression of several proteins, such as TGF-β or DNA methyltransferases (DNMTs) [5].

The Wnt/β-catenin signaling pathway plays a crucial role in maintaining intestinal epithelial homeostasis, regulating processes such as cell proliferation, differentiation, and migration [6]. The inactivation of β-catenin leads to a rapid loss of IECs, starting with crypt depletion, which coincides with blocked proliferation, increased enterocytes, and intestinal stem cell differentiation, resulting in compromised barrier function [7]. The dysregulation of this pathway is also linked to the development of colorectal cancer, where aberrant β-catenin activation drives uncontrolled cell growth.

Intestinal organoids have emerged as a powerful in vitro tool for studying intestinal biology due to their resemblance to in vivo tissue at the structural and functional levels [8]. Organoids are stem-cell-derived, self-organizing 3D structures supported by an extracellular matrix that contain multiple cell types of which the spatial arrangement, interactions, and functions mimic those of the native organ.

Our study investigates the effects of MDEs on β-catenin expression and related cellular behaviors in both healthy and cancerous colon epithelial cells, exploring their potential as therapeutic agents in both IBD and colorectal cancer. Given the promising results of MDEs in preclinical colitis models, we also examined their effects on patient-derived colon organoids (PDCOs). This model enables us to evaluate the translatability of MDEs’ therapeutic effects from animal models to human disease, providing critical insights into their clinical potential.

## 2. Results

### 2.1. The Isolation and Characterization of Milk-Derived Extracellular Vesicles

Extracellular vesicles were successfully isolated from cows’ milk and characterized by their morphology, size, protein content, and miRNA expression. Transmission electron microscopy (TEM) showed that the EVs exhibited a characteristic round- or cup-shaped morphology (Figure 1A) distinct from typical casein micelles or fat globules. The distribution size of the MDEs was determined using dynamic light scattering (DLS), which showed a Zeta-average (d-nm) diameter of 158.3 nm and a polydispersity index (PDI) of 0.1596 (Figure 1B), indicating a relatively uniform size distribution, as evidenced by a sharp, single main peak.

Western blot analysis confirmed the presence of CD81, a key exosome-associated protein, in the MDEs (Figure 1C). Additionally, the expression of several miRNAs, which are known cargo molecules within EVs, including exosomes, was analyzed, and their relative abundance is demonstrated in Figure 1D. Collectively, these results confirm the successful isolation of highly pure EVs from cows’ milk and their comprehensive characterization using multiple methods.

### 2.2. MDEs Induce β-Catenin Expression in a Murine Colitis Model Induced by DSS

Colitis was induced in Balb/c mice using 3.5% DSS provided for 6 days in their drinking water. Then, the mice were treated for 6 days with or without MDEs as a control. DSS-treated mice that did not receive MDEs showed significant weight loss starting on day 6 following DSS treatment (a decrease of 8.08%), with the greatest loss on day 8 (a decrease of 11.14%), continuing until the end of the experiment (a decrease of 6.05%). In contrast, DSS-treated mice that received MDEs showed significantly less weight loss only apparent on day 8 following DSS treatment (a decrease of only 5.27%) (Figure 2A). Moreover, MDE complementation induced a beneficial effect on the signs and symptoms of DSS-induced colitis. From day 5 to day 10, the disease score of the untreated mice was significantly higher compared to control mice (no DSS treatment). The MDE-treated mice only had a significantly higher score compared to control mice on day 6 (Figure 2B). The mean colon length of control mice was 9.8 ± 0.6 cm. Treatment with MDEs significantly reduced colon shortening induced by DSS treatment (*p* < 0.05) (Figure 2A). Following treatment with MDEs, the average length of the colon was 8.4 ± 0.6 cm compared with 6.6 ± 1 cm, which was the median length of the colon in the DSS group not treated with MDEs (Figure 2C). Representative histological images of colons from MDE-treated and untreated mice are shown in Appendix A.

In addition to these phenotypic observations, protein expression analysis provided further insights into the mechanistic effects of MDE treatment. There was a significant upregulation of β-catenin protein in the colonic tissues of the MDE-treated mice (Figure 2D). Quantification showed that β-catenin expression in the MDE-treated group was significantly elevated compared to the β-catenin expression in the untreated DSS colitis controls (Figure 2E).

### 2.3. MDEs Are Efficiently Uptaken by Patient-Derived Colon Organoids, Altering miRNA Profiles and mRNA Target Genes

Fluorescently labeled MDEs were incubated with PDCOs to assess their uptake and impact on miRNA and mRNA expression. Following incubation, fluorescence was observed within the PDCOs, confirming that MDEs successfully entered the organoids embedded in 50% Matrigel (Figure 3A). Subsequent RNA expression analysis revealed that the levels of two miRNAs, Let-7a and miR-148a, which are abundant in MDEs, were significantly elevated in the PDCOs after MDE incubation (Figure 3B). Furthermore, as expected, the expression of human ***DNMT1***, a known target gene of miR-148a, was downregulated in the PDCOs following MDE treatment (Figure 3C).

### 2.4. MDEs Upregulate β-Catenin, Cyclin D1, and the Proliferation Marker Ki67 in Patient-Derived Colon Organoids (PDCOs)

Given the observed upregulation of β-catenin in the colonic tissues of colitis-affected mice treated with MDEs, we extended our analysis to PDCOs derived from the colon tissues of IBD patients. As shown in Figure 4A, β-catenin expression was significantly upregulated in PDCOs following MDE treatment compared to Phosphate-buffered saline (PBS) controls. Consistent with β-catenin’s role in regulating cyclin D1, we also found a significant increase in *cyclin D1* mRNA levels in MDE-treated PDCOs (Figure 4B). Since β-catenin and cyclin D1 are key drivers of cellular proliferation, we further assessed IEC proliferation by examining the expression of the proliferation marker Ki67. Our results indicate that MDE treatment induced their proliferation, as evidenced by the upregulation of Ki67 (Figure 5).

### 2.5. MDEs Downregulate β-Catenin and Cyclin D1 and Reduce Proliferation in Colon Cancer Cells

In light of the observed upregulation of β-catenin and *cyclin D1* in PDCOs and colitis-affected mouse colon tissues, we investigated how MDEs influence these proteins in a cancer context. This analysis was crucial to determine whether the upregulation of these proteins could inadvertently stimulate the proliferation of malignant cells. Contrary to the upregulation observed in PDCOs, incubation of colon cancer cells with MDEs resulted in the downregulation of both β-catenin and cyclin D1 (Figure 6A). Additionally, MDE treatment led to a reduction in the proliferation rate of these cancer cells (Figure 6B).

## 3. Discussion

Our study demonstrated that MDEs not only ameliorate DSS-induced colitis but also upregulate β-catenin expression in colonic tissues. Extending our previous work in animal models, we confirmed these effects in PDCOs. MDEs were efficiently internalized in PDCOs, altering miRNA profiles and downregulating mRNA target genes, leading to increased expression of β-catenin, cyclin D1, and the proliferation marker Ki67. Interestingly, MDEs exhibited the opposite effect in colon cancer cells, downregulating β-catenin and cyclin D1 and reducing cell proliferation.

Previous studies have established the therapeutic potential of MDEs in colitis models. MDEs have been shown to reduce inflammation, mitigate weight loss, restore colon length, and decrease clinical signs of colitis in these models. These beneficial effects are primarily attributed to the anti-inflammatory properties of MDEs and to their ability to modulate immune responses and decrease pro-inflammatory cytokines such as TNFα [9,10]. While this study focused on epithelial cells, future research should explore MDEs’ effects on inflammatory cells, given their known anti-inflammatory properties, such as modulating cytokine expression and immune responses in IBD models. Furthermore, MDEs have been implicated in maintaining intestinal immune homeostasis via reshaping the gut microbiota, protecting against colitis by reversing dysbiosis [11,12].

In healthy individuals, the IECs play a role in maintaining immune homeostasis and maintaining interactions with the intestinal microbiota. Their barrier function and lineage-specific activities are critical to preventing chronic intestinal inflammation, underscoring their significance in the pathophysiology of IBD [13]. Direct damage to IECs, as caused by mucosal irritants or cytotoxic agents, results in a significant loss of barrier function.

Our findings align with previous reports suggesting that MDEs enhance intestinal epithelial barrier integrity by promoting IEC proliferation, differentiation, and biological function of IECs [4,14]. β-catenin, a key regulator of cell survival and proliferation, was upregulated in MDE-treated mice colons, possibly contributing to epithelial barrier repair. This regulation of the β-catenin pathway by MDEs was also observed in other models, including hair regeneration. MDEs accelerated the hair cycle transition from the telogen to anagen phase by activating the Wnt/β-catenin pathway and increasing the proliferation rate of Dermal Papilla Cells [15]. Furthermore, human MDEs were found to protect intestinal stem cells from oxidative stress injury in vitro, which was possibly mediated via the Wnt/β-catenin signaling pathway [16]. The translation of these findings to human models was performed in this study by using PDCOs. We showed that MDEs induced epithelial cell proliferation, further supporting their potential role in IEC repair. One of the mechanisms by which β-catenin induces proliferation of epithelial cells is by inducing genes such as cyclin D1. The expression of cyclin D1 that likely promotes cell proliferation is induced by β-catenin overexpression [17]. Our data demonstrate that MDE upregulates cyclin D1 expression in patient-derived colon organoids, aligning with previous studies in intestinal stem cells [16] and Saos-2 cells treated with MDEs [18].

The Wnt/β-catenin signaling pathway plays a crucial role in colorectal carcinogenesis [19], prompting us to investigate the effects of MDEs on cancer cells. In this study, we observed a significant downregulation of β-catenin and reduced proliferation in colon cancer cells treated with MDEs. This context-dependent effect contrasts with the proliferative impact of MDEs on normal cells, suggesting that MDEs selectively modulate β-catenin based on cell type. This selective mechanism may explain their distinct effects on cancerous versus non-cancerous cells [4], highlighting the therapeutic potential of MDEs to target healthy epithelial cells without promoting tumorigenesis. The differential regulation of β-catenin likely contributes to the cell type-specific effects exerted by MDEs. While our findings suggest a differential response of MDEs in inflammatory bowel disease and colon cancer models, we acknowledge that these conclusions are limited by differences in cell composition and culture types between the models used. Further studies incorporating multicellular spheroid cultures of LS123 cells or inducing inflammation, such as through LPS treatment, are recommended to better understand the context-dependent effects of MDEs.

## 4. Materials and Methods

Appendix A presents a graphical overview of the study design, summarizing the main experimental steps.

### 4.1. Milk Sample Collection

Cows’ milk was collected before pasteurization from a pool of milk. The samples were transported to the laboratory and stored at −80 °C for further processing.

### 4.2. Extracellular Vesicles Isolation from Milk

Extracellular vesicles were isolated through sequential ultracentrifugation and filtration. The milk samples were initially centrifuged at 3000× *g* for 30 min at 4 °C to separate the fat and skim milk fractions. EVs were extracted from the skim milk fraction, which was further centrifuged at 12,000× *g* for 1 h at 4 °C to remove debris. The resulting supernatant was filtered through 0.45 μm and 0.22 μm filters to remove residual debris. Subsequently, the filtrate was centrifugated at 135,000× *g* for 90 min at 4 °C to pellet the EVs. The pellet was resuspended in PBS and incubated overnight at 4 °C to dissolve the MDEs. The MDEs were then filtered using a 0.22 μm filter. The protein content of the EVs preparation was quantified using a Bicinchoninic Acid (BCA) protein assay.

### 4.3. Electron Microscopy

Isolated MDEs were diluted in PBS, vortexed for 3 min, and stained with 5% Uranyl Acetate. Following staining, the samples were air-dried. The stained MDEs were analyzed using electron microscopy as previously described [4], using a Jem-1400 Plus transmission electron microscope (Jeol, Peabody, MA, USA).

### 4.4. Dynamic Light Scattering

Dynamic light scattering (DLS) and zeta potential measurements were performed using a Zetasizer Nano series instrument (λ = 532 nm laser wavelength; Malvern Nano-Zetasizer, Worcestershire, UK). The size distribution of MDEs was based on scattering intensity (z-average).

### 4.5. Immunoblotting

MDEs were lysed in SDS, separated by SDS-PAGE (30 μg of protein), and transferred to a PVDF membrane. The membranes were probed with primary antibodies against CD81 (1:1000; Cosmo Bio, Tokyo, Japan). The secondary antibody used was horseradish peroxidase (HRP)-conjugated goat anti-mouse or anti-rabbit (1:3000; Cell Signaling Technology, Danvers, MA, USA).

### 4.6. RNA Extraction

Total RNA was extracted from the MDE pellet using Trizol reagent (INVITROGEN, Carlsbad, CA, USA), following the method described by Gutman et al. [20]. Cells and organoids were collected and suspended with TRIzol reagent for RNA extraction. RNA isolation was performed using the Zymo Direct-zol RNA MiniPrep Kit (Zymo Research, Irvine, CA, USA) according to the manufacturer’s instructions. RNA quantity and quality were assessed using a NanoDrop spectrophotometer (Waltham, MA, USA).

### 4.7. mRNA Detection by qRT-PCR

For the quantification of mRNA, complementary cDNA was synthesized from 1 µg of total RNA using the high-capacity RNA-to-cDNA kit (Applied Biosystems, Foster City, CA, USA) per the manufacturer’s instructions. mRNA levels were measured using quantitative real-time PCR (qRT-PCR) using Fast qPCR SYBR Green Blue Mix (PCR Biosynthesis, Wayne, PA, USA). The 2^−ΔΔCT^ method was used to calculate relative mRNA levels, with GAPDH as the reference gene. A complete list of primers is available in the Appendix A.

### 4.8. MicroRNA Detection by qRT-PCR

For miRNA analysis, 500 ng of total RNA extracted from cells and 100 ng of RNA extracted from MDEs were reverse transcribed using the qScript miRNA cDNA Synthesis Kit (Quantabio, Beverly, MA, USA), according to the manufacturer’s instructions. The resulting cDNA was used to measure the expression of miR-148a and Let-7a, using RNU6 as the reference gene. The PerfeCTa SYBR Green SuperMix (Quantabio, Beverly, MA, USA) was used together with Quantabio (Beverly, MA, USA) micro-RNA qPCR primers for the miR-148a-3p (HSMIR-0148A-3P), Let-7a (HSLET-0007A-5P), and RNU6 (HS-RNU6). The qRT-PCR was run using a two-step cycling protocol as previously described [21]. Normalization and relative expression levels were calculated using the 2^ (−ΔΔ CT) method.

### 4.9. Cell Cultures

LS123 colonic cancer cells and CCD 841 normal colon epithelial cells were cultured in Eagle’s Minimum Essential Medium (MEM) supplemented with 10% fetal calf serum (FCS), 100 U/mL penicillin, and 100 μg/mL streptomycin. Cells were incubated at 37 °C with 5% CO_2_.

### 4.10. Dextran Sulfate Sodium (DSS)-Induced Colitis Model in Mice

Colitis was induced in 8-week-old male Balb/c mice (Envigo RMS, Rehovot, Israel) by administering 3.5% DSS (MP Biomedicals, Illkirch, France) in drinking water for 6 days, resulting in colitis grade 36,000–50,000. To assess the effects of DSS, behavior and body weight were monitored daily. Starting on day 6, DSS was replaced with regular water. Mice received 50 mg/kg of cows’-milk-derived MDEs in 200 μL PBS orally by gavage for 6 days. At the end of the experiment, the mice were sacrificed, and their colons were excised, examined, and measured for length. After removing the cecum and adipose tissue, the colon samples were sectioned (proximal, middle, and distal). The proximal and middle sections were stored at −80 °C for protein expression analysis. Protein content was measured using a BCA protein assay (Thermo, Waltham, MA, USA). The experimental protocols were approved by the Ethics Committee (research number MD-20-15923-4).

### 4.11. Cell Viability and Apoptosis Assay

LS123 cells were incubated with 0.1 mg/mL of MDEs for 2, 6, and 24 h. Cell viability was measured using the Cell Counting Kit-8 (CCK8), according to the manufacturer’s instructions, with absorbance measured at 450 nm.

### 4.12. Patient-Derived Colon Organoids

Tissue samples were collected from participants undergoing colonic resection for inflammatory bowel disease. Control samples were obtained from non-diseased colonic tissue of patients undergoing colonic resection for colorectal cancer. All samples were collected with informed consent and in accordance with ethical guidelines (ethics approval number HMO-0921-20). Crypts were isolated from tissue samples by thoroughly washing biopsies in cold PBS to remove debris. The biopsies were then incubated in PBS supplemented with EDTA (final concentration 10 mM) to facilitate crypt detachment. Following EDTA incubation for 90 min, the crypts were released by vigorous pipetting and collected from the supernatant. This process was repeated until a sufficient quantity of crypts was obtained. Finally, the crypts were washed and centrifuged to concentrate the pellet for further use. The isolated crypts containing stem cells were then seeded in Matrigel in 24-well plates using a well-established protocol [22,23]. Vancomycin, gentamicin, and ROCK inhibitor (Y-27632) were added during the first week of culture. The growth medium was refreshed every 2–3 days, and organoids passed every 7–10 days. Bright-field images of patient-derived colon organoids have been included in the Appendix A to provide a visual representation of their morphology (Appendix A).

### 4.13. Labeling of MDEs

MDE RNA was labeled using acridine orange (Invitrogen, Carlsbad, CA, USA). We added 1 μL of acridine orange (10 mg/mL) to 100 μL of PBS containing sEVs derived from 5 × 10^5^ MSCs. After thorough mixing, the sample was incubated for 10 min at 37 °C. To halt the labeling process, 30 μL of ExoQuick-TC reagent (System Biosciences, Palo Alto, CA, USA) was added to the labeled MDE suspension, and the mixture was inverted six times. The sample was then placed on ice for half an hour, after which it was centrifuged at 1500× *g* for 3 min to sediment the vesicles. The pellet containing the labeled MDEs was resuspended in PBS and incubated with the organoids. Images were collected using a Nikon TL (Nikon Corporation, Tokyo, Japan).

### 4.14. Immunofluorescence Staining of Patient-Derived Colon Organoids

The organoids were fixed in 2% paraformaldehyde (PFA) for 20 min at room temperature (RT), permeabilized with 0.5% Triton X-100 in PBS, and blocked with 10% serum for 1–1.5 h. They were incubated overnight at 4 °C with primary antibodies against rabbit anti-β-catenin (1:100, Cell Signaling Technology, Danvers, MA, USA) and mouse anti-Ki67 (1:100, Cell Signaling Technology, Danvers, MA, USA. After washing, secondary antibodies (1:1000 goat anti-rabbit or anti-mouse IgG (Alexa Fluor 488, Abcam, Cambridge, MA, USA) were applied for 1 h in the dark. Following additional washes, the nuclei were stained with DAPI (Abcam, Cambridge, MA, USA), and the organoids were mounted for imaging. Images were collected using a Nikon TL (Nikon Corporation, Tokyo, Japan).

## 5. Conclusions

Our findings emphasize the context-specific effects of MDEs on β-catenin expression and cell behavior in both normal and cancerous colon epithelial cells. MDEs demonstrate therapeutic potential for both IBD and colorectal cancer. Future studies should further explore these differential effects across various cell types to optimize the therapeutic applications of MDEs.

## Figures and Tables

**Figure 1 ijms-25-13519-f001:**
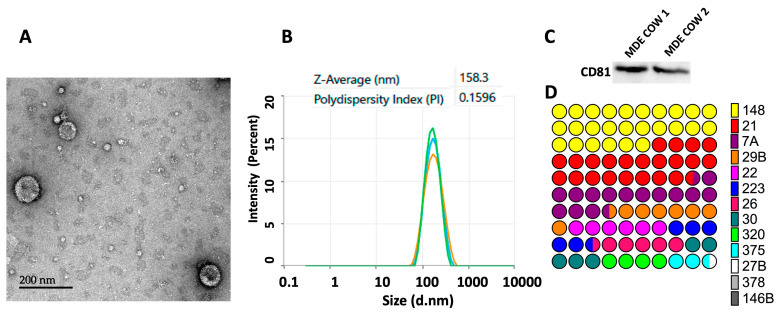
Characterization of MDEs. (**A**) Transmission electron microscopy (TEM) with negative staining was used to visualize MDEs. (**B**) The particle size distribution of isolated MDEs was determined by dynamic light scattering (DLS), with results performed in triplicate indicating an average diameter of 158.3 nm. (**C**) Western blot analysis was performed to assess the protein expression of Cluster of Differentiation 81 (CD81). (**D**) Quantitative real-time PCR (qRT-PCR) was used to measure the expression of abundant microRNA (miRNA) in MDEs. MiRNA expression was calculated using the 2^−ΔΔCT^ method and normalized to RNU6. miR-148 (148), miR-21 (21), Let-7a (7A), miR-29b (29B), miR-22 (22), miR223 (223), miR-26a (26), miR-30 (30), miR-320 (320), miR-375 (375), miR-27B (27B), miR-378 (378) and miR-146b (146B).

**Figure 2 ijms-25-13519-f002:**
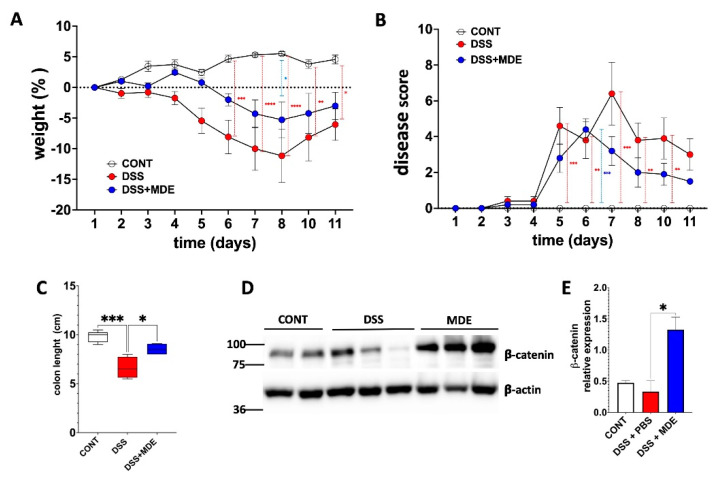
Effects of MDEs on a dextran sulfate sodium (DSS)-induced colitis model in mice. MDEs were orally administrated to mice with DSS-induced colitis. (**A**) Body weight of control (CONT) mice, colitic mice treated with (DSS + MDE), and untreated colitic mice (DSS). (**B**) Disease activity score in CONT, DSS + MDE, and DSS mice. (**C**) Colon length in control CONT, DSS + MDE, and DSS mice. (**D**) β-catenin protein expression in the colon of CONT, DSS + MDE, and PBS (DSS) mice. Protein expression was determined by Western blot, with β-actin as the loading control. (**E**) Quantification of β-catenin protein levels was performed using the ImageJ software, (U. S. National Institutes of Health, Bethesda, Maryland, USA, https://imagej.net/ij/). Data are presented as mean ± SEM, n = 8 per group, * *p* < 0.05, ** *p* < 0.01, *** *p* < 0.001, **** *p* < 0.0001 (Mann–Whitney test).

**Figure 3 ijms-25-13519-f003:**
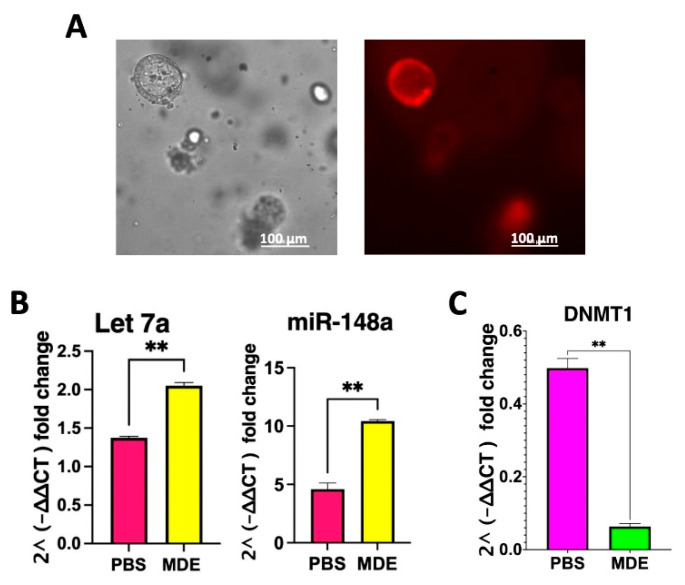
MDE uptake and miRNA expression in intestinal organoids. (**A**) Fluorescent images of 3D patient-derived colon organoid (PDCO) cultures incubated with labeled cow MDEs (0.1 mg/mL) were obtained by fluorescence microscopy. (**B**) Expression of let-7 and miRNA-148 in MDE-treated (0.1 mg/mL) (MDE) and untreated (PBS) PDCOs were assessed by qRT-PCR. MiRNA expression was calculated using the 2^−ΔΔCT^ method and normalized to RNU6. (**C**) *DNMT1* expression in MDE-treated (MDE) and untreated (PBS) PDCOs was analyzed by qRT-PCR. mRNA expression was calculated using the 2^−ΔΔCT^ method and normalized to GAPDH. Data are presented as mean ± SEM. ** *p* < 0.01. Scale bar: 100 μm.

**Figure 4 ijms-25-13519-f004:**
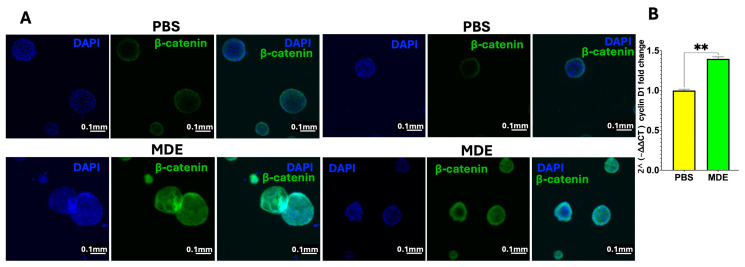
The effect of MDE treatment on β-catenin and *cyclin D1* expression in patient-derived colon organoids (PDCOs). (**A**) Immunostaining analysis of PDCOs from IBD patients treated with 0.1 mg/mL MDE (MDE) or untreated (PBS), showing β-catenin expression. Representative images from fluorescent microscopy. Scale bar: 0.1 mm. (**B**) Expression of cyclin D1 in MDE-treated (MDE) and untreated (PBS) PDCOs was analyzed by qRT-PCR. Values were calculated using the 2^−ΔΔCT^ method and normalized to *GAPDH*. Data are presented as mean ± SEM. ** *p* < 0.01.

**Figure 5 ijms-25-13519-f005:**
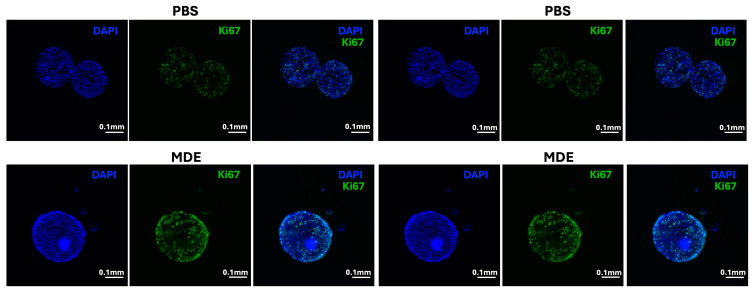
The effect of MDE treatment on proliferation of patient-derived colon organoids (PDCOs). Representative images from immunostaining analysis of PDCOs from IBD patients treated with 0.1 mg/mL MDE (MDE) or untreated (CONT) showing proliferative cells (Ki67). Images were acquired using fluorescent microscopy. Scale bar: 0.1 mm.

**Figure 6 ijms-25-13519-f006:**
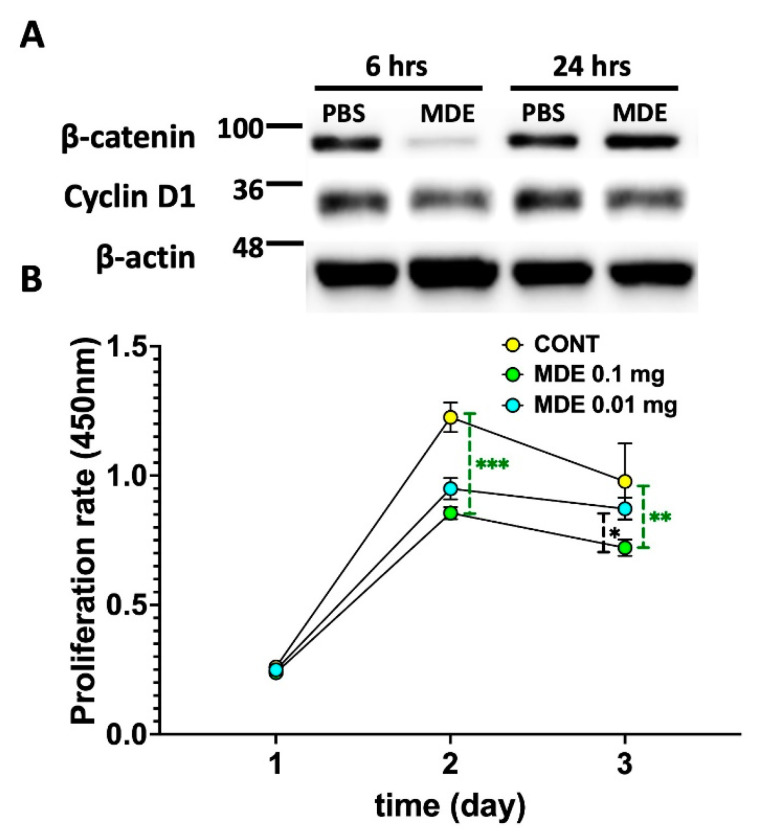
The effect of MDEs on human colon tumor cells (LS123). (**A**) Expression of β-catenin and cyclin D1 protein in LS123 cells incubated with MDEs at two different concentrations (0.1 and 0.01 mg/mL MDE) or without MDEs (CONT). Protein expression was determined by Western blot, with β-actin used as the loading control. (**B**) LS123 cells grown in 0% fetal calf serum (FCS) and incubated with MDEs at two different concentrations (0.1 and 0.01 mg/mL MDE) or without (CONT) were examined using the CCK-8 assay to assess cell growth. Error bars represent SD (n = 4). * *p* < 0.05, ** *p* < 0.01, *** *p* < 0.001.

## Data Availability

The raw data supporting the conclusions of this article will be made available by the authors on request.

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
