# Peer review of "The Effect of Milk-Derived Extracellular Vesicles on Intestinal Epithelial Cell Proliferation"

_ijms, 2024, doi:10.3390/ijms252413519_

Round 1

Reviewer 1 Report

Comments and Suggestions for Authors

The authors of the submitted article describe the effect of milk-derived extracellular vesicles on epithelial cells of inflammatory bowel disease and on LS123 colon cancer cells.

Comments

The EVs are sufficiently characterized.

Gut histology of the DSS-induced treated and untreated mice should be shown.

Materials and methods need to be described in more detail regarding the following points:

·         Fluorescence labeling of the EVs

·         Fixation and staining of MEDs for TEM

·         Isolation of the crypts, umber of tissues samples collected from patients

·         Provider of the antibodies, type of secondary antibody, dilutions of antibodies

·         Instrumentation and setting of imaging

Green labeling is not well visible in Figs 4 and 5 – better show images as separate channels and merged.

The conclusion that the EVs react differently in inflammatory bowel disease and colon cancer is not fully justified by the data because cell composition and culture type are different. Induction of inflammation (e.g. LPS treatment) and/or culture as multicellular spheroids of L123 cells is needed for valid comparison.

Minor

Scale bar in the Figures 4 and 5 (0.1 nm) is not correct

Author Response

Reviewer 1

Thank you for your thorough review and valuable suggestions, which have greatly helped to improve the quality and clarity of our manuscript. Below, we provide detailed responses to your comments and outline the revisions made.

Comments and Suggestions for Authors

The authors of the submitted article describe the effect of milk-derived extracellular vesicles on epithelial cells of inflammatory bowel disease and on LS123 colon cancer cells.

Comments

The EVs are sufficiently characterized.

  • Gut histology of the DSS-induced treated and untreated mice should be shown.

Response:We have added histological images of the gut from DSS-induced treated and untreated mice in the supplementary results section, as suggested (Supplementary Figure 1).

  • Materials and methods need to be described in more detail regarding the following points:
  • Fluorescence labeling of the EVs

Response: Details about the fluorescence labeling protocol for the EVs have been included in the "Materials and Methods" section (Subsection 4.13: Labeling of MDEs).

  • Fixation and staining of MEDs for TEM

Response: We have expanded the description of the fixation and staining protocol for TEM analysis in the "Materials and Methods" section (Subsection 4.3: Electron Microscopy).

  • Isolation of the crypts, umber of tissues samples collected from patients

Response: Thank you for your valuable comment. We have now clarified the origin of the tissue samples and provided additional details regarding the crypt isolation process.Details about the crypt isolation procedure and the number of tissue samples collected from patients have been clarified in the "Materials and Methods" section (Subsection 4.12: Patient-derived colon organoids).

  • Provider of the antibodies, type of secondary antibody, dilutions of antibodies.

Response: We have specified the type of secondary antibodies, and antibody dilutions in the "Materials and Methods" section (Subsection 4.14: Immunofluorescence staining).

  • Instrumentation and setting of imaging

Response: Information on the instrumentation and imaging settings has been provided in the "Materials and Methods" section (Subsections 4.13 and 4.14).

  • Green labeling is not well visible in Figs 4 and 5 – better show images as separate channels and merged.

Response:We have revised Figures 4 and 5 to include images as separate channels and merged, improving visibility as per your recommendation.

  • The conclusion that the EVs react differently in inflammatory bowel disease and colon cancer is not fully justified by the data because cell composition and culture type are different. Induction of inflammation (e.g. LPS treatment) and/or culture as multicellular spheroids of L123 cells is needed for valid comparison.

Response: Thank you for this important observation. We have added a discussion in the "Discussion" section addressing how differences in cell composition and culture type may influence the differential responses observed. While we agree that LPS treatment or multicellular spheroid cultures of LS123 cells would provide valuable insights, this was beyond the scope of the current study. We have acknowledged this limitation and proposed these approaches as future directions for follow-up studies

Minor

Scale bar in the Figures 4 and 5 (0.1 nm) is not correct

Response:We have corrected the scale bar in Figures 4 and 5 to reflect the appropriate unit of measurement.

Reviewer 2 Report

Comments and Suggestions for Authors

Manuscript entitled "The effect of milk-derived extracellular vesicles (MDEs) on intestinal epithelial cells proliferation"

Major issues:

1. The authors should provide a graphic illustration of the study design.

2. For the patient-derived organoid, the patients should provide the IRB approval and the stained figures to check morphology under the light microscope.

3. Since IBD contains an interaction of inflammatory cells and epithelial and stromal cells. The authors should provide the effects of MDE on inflammatory cells.

4. The content of MDE should be explored using high-throughput study including LC-MS/MS and NGS.

5. For a more comprehensive study, an animal model would be suggested to be performed.

Author Response

Reiewer 2

We sincerely appreciate your constructive feedback and insightful comments, which have allowed us to address key aspects of the study. Please find our detailed responses and the corresponding revisions below

Comments and Suggestions for Authors

Manuscript entitled "The effect of milk-derived extracellular vesicles (MDEs) on intestinal epithelial cells proliferation"

Major issues:

  1. The authors should provide a graphic illustration of the study design.

Response: Thank you for this suggestion. A graphic illustration of the study design has been created and is now included in the manuscript as Supplementary figure 2.

  1. For the patient-derived organoid, the patients should provide the IRB approval and the stained figures to check morphology under the light microscope.

Response: Thank you for these insightful comments. The IRB approval number for the patient-derived colon organoids is provided in the Methods section. Additionally, we have included bright-field images of the organoids in the supplemental data to illustrate their morphology (Supplementary Figure 3).

  1. Since IBD contains an interaction of inflammatory cells and epithelial and stromal cells. The authors should provide the effects of MDE on inflammatory cells.

Response: Thank you for your valuable comment. We agree that the interaction between inflammatory cells, epithelial cells, and stromal cells is a critical component of IBD pathophysiology. While the primary focus of our current study was on the effects of MDEs on epithelial cells, we acknowledge the importance of understanding their impact on inflammatory cells. Previous studies, including our own work, have demonstrated that MDEs exhibit anti-inflammatory properties, such as modulating cytokine expression and reshaping immune responses. For example, it has been shown that MDEs can reduce levels of pro-inflammatory cytokines like TNFα and IL-6 while promoting anti-inflammatory cytokines (e.g., IL-10) in various inflammatory models (Reif et al., 2020; Tong et al., 2021).

In the current study, although we did not directly assess the effects of MDEs on inflammatory cells, we propose this as a key avenue for future research. Such studies could involve co-culture systems of epithelial and immune cells, or in vivo experiments evaluating the recruitment and activity of immune cells during MDE treatment in colitis models. These approaches will allow us to further elucidate the mechanisms by which MDEs modulate the IBD microenvironment.

To address this point in the revised manuscript, we have included a discussion highlighting the potential role of MDEs in influencing inflammatory cells and suggesting this as a direction for future studies.

  1. The content of MDE should be explored using high-throughput study including LC-MS/MS and NGS.

Response: Thank you for your comment. The content of MDEs has already been explored in our previous studies using high-throughput methods such NGS, as referenced in the manuscript. These analyses provided a comprehensive characterization of MDE cargo, which served as the foundation for the functional studies presented in this work. As such, these analyses are not part of the current study.

  1. For a more comprehensive study, an animal model would be suggested to be performed.

Response: Thank you for your suggestion. While this study does include an in vivo DSS-induced colitis model, the focus was on understanding the expression of beta catenin in colon tissue in vivo. These results provide a mechanistic basis for the observed effects of MDEs. Further animal studies exploring additional pathways or conditions could complement these findings, but they are beyond the scope of this current work.

Round 2

Reviewer 1 Report

Comments and Suggestions for Authors

My comments have been addressed (The incorrect indication of the scale bar was in the image itself (0.1 nm), not in the legend)

Reviewer 2 Report

Comments and Suggestions for Authors

The revision is acceptable for publication.